# Inhibition of Clinical MRSA Isolates by Coagulase Negative Staphylococci of Human Origin

**DOI:** 10.3390/antibiotics13040338

**Published:** 2024-04-08

**Authors:** Ellen Twomey, Paula M. O’Connor, Aidan Coffey, Maija Kiste, Caitriona M. Guinane, Colin Hill, Des Field, Máire Begley

**Affiliations:** 1Department of Biological Sciences, Munster Technological University, T12 P928 Cork, Ireland; etwomey@ucc.ie (E.T.); aidan.coffey@mtu.ie (A.C.); caitriona.guinane@mtu.ie (C.M.G.); 2APC Microbiome Ireland, University College Cork, T12 YN60 Cork, Ireland; c.hill@ucc.ie; 3Teagasc, Moorepark, Fermoy, P61 C996 Cork, Ireland; paula.oconnor@teagasc.ie; 4School of Microbiology, University College Cork, T12 YN60 Cork, Ireland

**Keywords:** bacteriocin, antibacterial peptide, purification, skin microbiome, *Staphylococcus epidermidis*, MRSA

## Abstract

*Staphylococcus aureus* is frequently highlighted as a priority for novel drug research due to its pathogenicity and ability to develop antibiotic resistance. Coagulase-negative staphylococci (CoNS) are resident flora of the skin and nares. Previous studies have confirmed their ability to kill and prevent colonization by *S. aureus* through the production of bioactive substances. This study screened a bank of 37 CoNS for their ability to inhibit the growth of methicillin-resistant *S. aureus* (MRSA). Deferred antagonism assays, growth curves, and antibiofilm testing performed with the cell-free supernatant derived from overnight CoNS cultures indicated antimicrobial and antibiofilm effects against MRSA indicators. Whole genome sequencing and BAGEL4 analysis of 11 CoNS isolates shortlisted for the inhibitory effects they displayed against MRSA led to the identification of two strains possessing complete putative bacteriocin operons. The operons were predicted to encode a nukacin variant and a novel epilancin variant. From this point, strains *Staphylococcus hominis* C14 and *Staphylococcus epidermidis* C33 became the focus of the investigation. Through HPLC, a peptide identical to previously characterized nukacin KQU-131 and a novel epilancin variant were isolated from cultures of C14 and C33, respectively. Mass spectrometry confirmed the presence of each peptide in the active fractions. Spot-on-lawn assays demonstrated both bacteriocins could inhibit the growth of an MRSA indicator. The identification of natural products with clinically relevant activity is important in today’s climate of escalating antimicrobial resistance and a depleting antibiotic pipeline. These findings also highlight the prospective role CoNS may play as a source of bioactive substances with activity against critical pathogens.

## 1. Introduction

*Staphylococcus aureus* is considered one of the most prominent bacterial pathogens of the 21st century. Responsible for chronic and recurrent wound infections, nasal infections, soft tissue infections, and atopic dermatitis [1,2,3,4], this organism is also widely recognized for its ability to develop resistance to antibiotics. The detection of *S. aureus* strains with resistance to critical last line of defense drugs, is a common occurrence. A report produced by the World Health Organization (WHO) concerning levels of resistant bacteria in US hospital settings suggests that approximately 90% of clinical *S. aureus* isolates are resistant to antibiotics [5]. This, coupled with statistics generated in 2019 that purport *S. aureus* may be responsible for up to 100,000 deaths worldwide [6], paints a clear picture of the growing threat this species presents to human health and wellbeing. It is precisely for these reasons that *S. aureus* is routinely flagged by leading healthcare bodies as a priority target for the upkeep of good hygiene practices and drug discovery research [7,8,9].

Although members of the same genus, Coagulase Negative Staphylococci (CoNS) are considered to be less pathogenic than *S. aureus* and are common constituents of the human commensal flora. CoNS are reported to be producers of an array of bioactive substances, including some that can exert antimicrobial effects against their coagulase positive counterpart e.g. proteases, quorum-sensing agents and toxins [10,11]. Iwase et al. found that purified serine proteases (Esp) produced by *Staphylococcus epidermidis* could inhibit the growth of *S. aureus* and destroy pre-formed biofilm [10]. Expanding on this finding through experimentation in murine models, Iwase et al. also demonstrated that intranasal application of Esp-producing *S. epidermidis* strains could make the host more resistant to colonization by methicillin-resistant *S. aureus* (MRSA) and could inhibit the growth of MRSA in a previously colonized host [12]. Phenol-soluble modulins (PSMs) are staphylococcal toxins with a narrow spectrum of activity [11]. When purified, a study by Cogan et al. purported that PSMs interfered with the *S. aureus* cell membrane and were effective at inhibiting the growth of MRSA at concentrations between 16 and 32 μM [11]. 

Of the many potential antimicrobials produced by CoNS, we focused on bacteriocins, small, ribosomally synthesized peptides with targeted activity against other closely related strains. These peptides have been confirmed to be active at nanomolar concentrations and do not appear to cause inflammation or cytotoxic effects when used *in vivo* [13]. Bacteriocins have been purified from a variety of different CoNS species and found to inhibit the growth of human pathogens, including MRSA. Well diffusion assays performed with nisin J, a naturally occurring nisin variant purified from human-isolated *S. capitis* APC 2923, caused inhibition of MRSA DPC 5645 [14]. In addition to this, diffusion assays found that nisin J appeared more potent than the previously characterized nisin A peptide, as the zone produced in the growth of the indicator was significantly larger [14]. This is noteworthy as nisin A is currently applied as a preservative in a variety of food products [14]. Epidermicin NI01, isolated from *S. epidermidis* 224, has also demonstrated anti-MRSA activity [15]. When applied to the nares of cotton rats colonized with MRSA ATCC 43300, a 0.8% solution of epidermicin NI01 suspended in hydroxypropyl methylcellulose significantly reduced or completely eradicated carriage of the pathogen [15]. Upon examination, no swelling or other harmful effects were observed as a result of using this peptide suspension, and the activity was described as comparable to that of mupirocin, a topical treatment effective against staphylococci currently available on the market [15]. The potency of CoNS-derived bacteriocins against MRSA, a serious human pathogen, makes them attractive options for the control of staphylococcal-associated infections. 

Despite decades spent developing novel iterations of pre-existing antibiotics to side-step reported resistance development and curb the spread of this species, resistant *S. aureus* strains continue to emerge to the point where isolates with multidrug resistance have become a commonplace occurrence worldwide [16,17,18]. Due to the urgent need to identify new treatments to manage MRSA and the evidence of bioactivity from CoNS in the literature, this study aimed to investigate the potential of a bank human-derived CoNS for their ability to inhibit the growth of clinical isolates of MRSA in vitro, and to discern if the production of bacteriocins may be responsible. 

## 2. Results

### 2.1. Identification of CoNS Isolates with Potential Antimicrobial Activity against MRSA 

As part of a large screen performed in our laboratory, CoNS isolates of human origin were shown to inhibit the growth of *Micrococcus luteus* through agar-based deferred antagonism assays (DAA) (M. Begley; unpublished data). Isolates that inhibited the growth of the indicator were stocked to create a biobank. From this bank, 37 CoNS isolates were selected for the current study. DAAs were repeated to confirm the antimicrobial activity, with all 37 CoNS confirmed to produce zones of clearing in the growth of *M. luteus*. Zone sizes were measured and ranged from 7 mm to 34 mm. The results of the DAAs are presented in Table 1. 

DAAs were then used to examine all 37 isolates for antimicrobial activity against 21 MRSA strains provided by the MTU Culture Collection, originally obtained from patients with infections. Nine CoNS isolates (C4, C5, C9, C14, C15, E54, E67, E109, and E170), produced zones of clearing in the growth of at least one MRSA indicator strain, with all zones of inhibition observed measuring <7 mm in diameter (see Table 1). Although only a few of the CoNS showed activity against MRSA, the high degree of bioactivity observed against *M. luteus* suggested all isolates warranted further investigation. An example of the inhibitory activity observed is presented in Figure 1 and depicts a representative isolate from the bank of CoNS with activity against *M. luteus* and a strain of MRSA. 

Well diffusion assays (WDAs) were performed with the acid-neutralized cell-free supernatant (CFS) prepared from overnight cultures of the CoNS isolates against the 21 aforementioned MRSA indicator strains and *M. luteus*. However, no zones of clearing were observed in the growth of any of the indicators following incubation.

### 2.2. Examination of the Ability of MRSA to Form Biofilms

In preparation for conducting broth-based antagonism and biofilm inhibition assays, the ability of each individual MRSA indicator strain to form an extracellular matrix was assessed as per the method previously established by Mathur et al. [19]. Three strong biofilm-forming MRSA strains (MRSA V, MRSA R, and MRSA E) and two moderate biofilm-forming strains (MRSA M and MRSA T) were selected for further use. 

### 2.3. Determination of Direct Antagonism against MRSA by CoNS

To investigate if the acid-neutralized CFS derived from CoNS isolate overnight cultures could impact the growth of MRSA in a liquid medium, the growth of a representative MRSA strain (MRSA M) cultured in the presence of CFS was monitored over a 12-h period. In the agar-based DAA, only CoNS isolate E67 produced a zone of inhibition against MRSA M (see Table 1). However, analysis of MRSA M absorbance values at T12 clearly indicated that all 37 isolates impacted the growth of this strain (see Appendix A). In the case of 36 of the 37 isolates, the average final absorbance value for MRSA M cultured in the presence of CFS was significantly reduced (*p* < 0.05) when compared to the positive control. Figure 2 presents the absorbances of indicator MRSA M cultured with the CFS of three representative strains over a twelve-hour period. 

### 2.4. Examination of Potential AntibiofilmEeffects within the Bank of CoNS Isolates

As the acid-neutralized CFS prepared from all 37 CoNS, overnight cultures appeared to have antagonistic effects against the MRSA M indicator strain, the potential bioactivity of the CoNS isolates were investigated further through an antibiofilm assay. Five MRSA indicator strains determined to be moderate or strong biofilm formers were incubated in the presence of the acid-neutralized CFS of each CoNS individually for 24 h under stationary conditions. The addition of the CoNS-derived CFS was seen to significantly reduce the absorbance readings obtained from the MRSA test wells when compared to the controls (MRSA indicators cultured in the absence of acid-neutralized CFS), with the CFS of each individual CoNS isolate observed to cause a significant reduction (*p* < 0.05) in biofilm recovered in at least two of the five indicators used (see Figure 3a–e). The CFS of 10 CoNS isolates (C28, C35, C41, C42, E3, E67, E75, E89, E100, and E109) were found to be capable of causing a statistically significant reduction in biofilm formed by all 5 MRSA strains, with the CFS of another 19 CoNS significantly inhibiting biofilm formation by four of the five MRSA strains.

### 2.5. Analysis of Whole Genome Sequencing of Shortlisted CoNS Isolates 

Eleven CoNS isolates (C14, C33, E4, E6, E54, E67, E89, E96, E99, E100, and E170) were selected for whole genome sequencing (WGS) based on the inhibitory activity observed against MRSA through DAAs, a growth inhibition assay, and biofilm inhibition assays. A summary of this activity observed is presented in Appendix A. Draft genomes were assembled for analysis. The results of the quality checks performed with QUAST and CheckM, sequence coverage determined with Bowtie2 and SAMtools, and the number of contigs are presented in Appendix A. The identities of the sequenced CoNS isolates were reassessed by aligning the 16S rRNA genes identified in the draft genomes with those possessed by members of the same species present on the NCBI RefSeq Database using a 97% identity match as the cutoff criteria. The species of all CoNS isolates aligned with the results obtained through prior partial 16s rRNA gene sequencing and which are listed in Section 4 Materials and Methods.

Analysis performed with the genome mining tool BAGEL4 detected areas of interest (AOIs) associated with the production of autoinducing peptides (AIPs), quorum sensing agents noted for interfering with the growth of closely related strains [20,21], in the genomes of all 11 CoNS isolates. Further inspection of these genomic regions and gene clusters through Artemis and BLASTp also identified genes encoding putative δ-lysins and phenol soluble modulins (PSMs) within each of the 11 sequenced genomes. δ-lysins and PSMs are small, *Staphylococcus*-associated bioactive peptides [11,22,23] that have previously been identified in human-derived CoNS and observed to exert antimicrobial activity against some gram-positive pathogens [11,24]. A summary of all AOIs identified in the preliminary investigation of the sequenced genomes using BAGEL4 is presented in Table 2. 

In addition to genes for the synthesis these lytic peptides, BAGEL4 identified AOIs associated with the production of a number of bacteriocins. In CoNS strains *S. capitis* E4, *S. capitis* E96 and *S. capitis* E170, operons potentially encoding a class I gallidermin family lantibiotic were detected. When aligned with the amino acid sequence of the previously reported gallidermin peptide [25], it appeared that the sequences of the putative core peptides within the suspected gallidermin operons identified in isolates E4, E96, and E170 were shorter than their characterized counterpart (see Figure 4b). Upon further inspection of the BAGEL4 report and the AOI presented in Artemis, it was also noted that the putative bacteriocin operons detected in the *S. capitis* isolates E4, E96, and E170 lacked accessory genes required for the production of an active peptide [25], i.e., immunity, modification, and export (see Figure 4a). This lack of genes critical for the biosynthesis of a bacteriocin and the shortened core genes suggest that the putative gallidermin-like peptides were likely neither synthesized nor active. On these grounds, isolates E4, E96, and E170 were disregarded from further investigation. 

Isolate C14 was confirmed to be a strain of *Staphylococcus hominis*, a CoNS species associated with antimicrobial peptide production [26,27], through analysis of the 16s rRNA genes. Inspection of the C14 genome using BAGEL4 identified a putative bacteriocin operon. Further analysis of the AOI using BLASTp indicated the operon may synthesize a nukacin family lantibiotic. Alignment of the amino acid sequence derived from the gene predicted to encode the core peptide identified in isolate C14 to those of other nukacin family bacteriocins reported on the Bactibase database and in the literature found that the sequence was a 100% identity match to that of nukacin KQU-131 (see Figure 5b). The nukacin KQU-131 peptide was previously purified from a strain of *S. hominis* isolated from Thai fermented fish [27]. As seen in Figure 5a, when compared to the previously characterized operon, the gene cluster identified in *S. hominis* C14 appears to possess all the necessary biosynthetic machinery for bacteriocin synthesis. For clarity within this study, the bacteriocin potentially produced by *S. hominis* C14 will be referred to as nukacin C14. 

A putative bacteriocin operon was also detected by BAGEL4 in the genome of strain C33 (identified as *S. epidermidis*). BLASTp analysis of each gene making up the cluster suggested that all the elements required for the synthesis of an epilancin family bacteriocin may be present. When the amino acid sequence of the gene predicted to encode the core peptide was aligned with that of previously characterized epilancin peptides, the sequence displayed high degree of similarity to that of epilancin 15X (81.8% amino acid identity match) [28] (see Figure 6b). When reviewing the literature, it was found that the specific bacteriocin precursor gene detected in isolate C33 has previously been identified in two separate *S. epidermidis* strains, *S. epidermidis* APC 3775 and *S. epidermidis* APC 3810 isolated as part of a breast milk screening study conducted by Angelopoulou et al. [29]. Although detected in silico, there have been no attempts to purify the peptide or characterize its activity [29]. 

We speculate that the putative bacteriocin operon detected in isolate C33 may be composed of 10 genes: a core peptide encoding gene, two genes associated with modification and processing, two biosynthesis-associated genes, one transport gene, and four genes associated with immunity. The genome had a Phred score of 36 (see Appendix A), and each gene analyzed had a high degree of similarity to those of the previously characterized epilancin 15X operon and the uncharacterized epilancin variant operons detected by Angelopoulou et al. [29]. Comparison of the potential epilancin encoding operon in isolate C33 to that of previously characterized epilancin 15X is presented in Figure 6a. Within this study, the uncharacterized epilancin variant potentially produced by *S. epidermidis* C33 will be referred to as epilancin E.

Pre-peptide epilancin family bacteriocins, epilancin 15X and epilancin, are both reported to be 55 amino acids long, with cleavage occurring for both at position 21, releasing a 34-amino-acid-long mature peptide. Due to the high percentage identity match to the amino acid sequences of both the epilancin 15X and epilancin K7 core peptides (amino acid identity match > 80%), the cleavage point of the leader sequence and length of the mature epilancin E peptide was predicted to be the same as its previously characterized homologs. The resulting amino acid sequence of mature epilancin E was then aligned with those of K7 and 15X and was predicted to differ at five and six separate positions respectively. Although the sequences differ, analysis with CLUSTAL W in MEGA11 indicated that the residues composing the novel epilancin variant shared the same biochemical and physicochemical properties as those of the 15X and K7 sequences. This suggests that despite differences in sequence, the final structure of the novel derivative likely corresponds to those of the other epilancin family bacteriocins. 

### 2.6. Purification and Analysis of the Bacteriocins Produced by the Human-Derived CoNS

Prior to purification, cultures of C14 and C33 were prepared in order to investigate if any bioactive substances were bound to the cell surface, as no activity had been seen previously when WDAs were performed with the CFS. This is a phenomenon reported with some bacteriocin-producing strains, whereby the peptides can be stripped from the cell surface through a combination of incubation in the presence of solvents such as isopropyl alcohol (IPA), and agitation. Despite investigating several different solvents, no activity was observed when WDAs were performed with the whole cell extract. As it was suspected that the amount of culture might impact the ability to visualize activity, the volume of culture was increased and a commonly used method in our laboratory for the isolation of the nisin peptide was employed in an attempt to confirm that the bacteriocins identified in strains C14 and C33 were produced and active against MRSA. 

HPLC was used to isolate both the nukacin C14 and epilancin E peptides. In the case of nukacin C14, two fractions were found to inhibit the growth of the sensitive indicator, *M. luteus*. MALDI TOF MS confirmed the presence of a mass peak correlating with that previously reported for nukacin KQU-131 [27] within the active fractions, indicating that the nukacin C14 peptide was being synthesized and may be responsible for the activity observed against MRSA. The HPLC profile, MALDI TOF MS reading and a WDA indicating the activity observed from the purified nukacin C14 peptide are presented in Figure 7.

In the case of epilancin E, HPLC separation produced eight fractions with activity against a sensitive indicator, *Lactococcus lactis* HP (fractions 48 through 55). MALDI-TOF MS analysis of the active fractions found that each possessed a peak at 2982 Da. The mature epilancin variant detected in the genome of *S. epidermidis* C33 was predicted to be 31 residues long with an amino acid sequence of ‘SASVVKTTVKASKKLCKGATLTCGCNITGKK’ and a mass of 3126 Da. The difference between the predicted mass and the mass detected in the active fractions correlates with the post-translational modifications observed in other epilancin family bacteriocins, specifically the dehydration of serine and threonine residues and the presence of cysteine residues leading to the formation of thioester rings [30]. The formation of the mature epilancin 15X and K7 peptides requires eight dehydration reactions in total (one lanthionine residue, two 3-methyllanthionine bridges, one 3-dehydroalanine residue, three (Z)-2,3-dehydrobutyrine residues, and an N-terminal lactate) [30]. These dehydrations equate to 144 Da, the same as the difference observed between the predicted mass and the mass of the active peptide detected post-purification [30]. The purified peptide was tested for its activity against *M. luteus* using WDAs. The HPLC profile with the active fractions highlighted in red, the MALDI TOF MS analysis with the mass of the active epilancin E peptide, and the results of the WDA against *M. luteus* are presented in Figure 8.

Following the isolation of nukacin C14 and epilancin E, the peptides were tested against representative MRSA indicators, MRSA M and MRSA T, using spot-on-lawn assays. This assay was selected as the concentration of both peptides was unknown, and diffusion through a solid medium may have impacted the ability to visualize activity. Following incubation with the indicators, it was observed that nukacin C14 caused zones of clearing in the growth of MRSA M and MRSA T, and epilancin E caused a zone of clearing in the growth of MRSA M. Representative spot on lawn assays to showcase the activity of the purified peptides against MRSA are seen in Figure 9. 

These findings suggest that the bacteriocins produced by the human-derived CoNS isolates may have had a role in suppressing the growth of MRSA and that CoNS-derived bacteriocins may be an effective treatment against this pathogen in the future. 

## 3. Discussion

CoNS are common colonizers of the human skin and nares. Along with exerting some probiotic effects on the skin through quorum sensing [20,21], interactions with the host immune system [31,32,33], and the secretion of antimicrobial enzymes [34,35], human-derived CoNS have been characterized as the producers of a variety of bacteriocins which when purified, have expressed clinically relevant bioactivity [14,36,37]. Given the relatedness of CoNS to their coagulase-positive counterparts and the targeted nature of bacteriocins, it is logical that CoNS-derived bacteriocins could be applied as a treatment against *S. aureus*. This study screened a bank of CoNS against clinical isolates of MRSA in an attempt to identify antimicrobial producers with activity against this specific pathogen. Twenty-one MRSA isolates were used to reduce the likelihood that natural variations in antimicrobial susceptibility amongst the *S. aureus* strains would result in a putative antimicrobial producer being overlooked. Unlike many other bacteriocin investigations, which screen CoNS against a broad range of indicators or a specific sensitive indicator during the preliminary testing [36,38,39,40], this investigation took a more directed approach with a single target pathogen in mind. 

Through the initial screening process of DAAs, all CoNS isolates were carried forward for further investigation due to the high degree of inhibitory activity observed within the bank. While approximately 24% of CoNS isolates prevented the growth of at least one MRSA indicator, all 37 caused large zones in the growth of *M. luteus*. A similar investigation by Lynch et al., which also sought to evaluate the antimicrobial potential of CoNS, found that 94 of 100 isolates within a bank could inhibit the growth of at least one of the 24 indicators utilized [36]. Again, a separate investigation of bioactivity undertaken by Janek et al. noted a high degree of inhibition by a bank of 89 CoNS isolates obtained from the nares of healthy volunteers. In this instance, 75 isolates (84% of the bank) prevented the growth of at least one indicator examined [39]. However, this success rate may be dependent on the study design rather than the specific species or genus tested. A large-scale study performed by O’Sullivan et al. observed a very small percentage of bioactivity using the same assay [38]. Over 90,000 CoNS isolates were screened against *Lactobacillus delbrueckii* ssp. *bulgaricus* LMG 6901, *Listeria innocua* DPC 3572, and MRSA DPC 5645 using agar-based DAAs and WDAs to identify potential antimicrobial producers [38]. Based on the activity observed during this screen, 21 isolates from the bank were shortlisted for further investigation. When compared to the initial size of the bank, the number of shortlisted isolates (0.023%) is extremely low [38]. This could potentially be due to the selection of indicator strains (choosing strains not closely related to the species of the isolates investigated) or using only one strain of each indicator selected (failing to consider strain-to-strain variances in susceptibility). In the present study, the inclusion of an appropriate sensitive indicator (*M. luteus*) allowed for the visualization of activity from all 37 CoNS. In addition, the issue of strain sensitivity variances was combated by using 21 separate MRSA indicators. To reaffirm the result and explore the anti-MRSA potential of the bank of CoNS, broth-based inhibition assays were attempted. 

Transitioning to a broth-based assay improved the amount of anti-MRSA activity observed within the bank. The acid-neutralized CFS of all isolates under investigation caused an observable decline in the growth of the selected indicator strain (MRSA M), with 36 causing a significant reduction in the OD600nm recorded at T12 when compared to the control (*p* < 0.05). This shift in observed antagonism was drastic and highlights the limitations of agar-based assays. Few large-scale broth-based antagonism assays performed using CoNS-derived CFS against MRSA are available in the literature for comparison. However, smaller studies performed in the same manner correlate with the degree of inhibition observed. Using a method akin to that of the present study, Chin et al. demonstrated that when incubated with *S. aureus*, the CFS of a CoNS isolate (*Staphylococcus chromogenes* ATCC43764) could significantly reduce the absorbance obtained at T24 [41]. CoNS-derived CFS has also been found to significantly reduce the formation of *S. aureus* biofilm when incubated with the culture [10]. Iwase et al. prepared CFS from 16-h-old cultures of *S. epidermidis* obtained from the nares of 88 volunteers. It was then observed that of 960 *S. epidermidis* isolates investigated, the CFS of 428 could reduce *S. aureus* biofilm formation in a dose-dependent manner [10]. The present study reports similar antibiofilm-forming effects within the bank, as significantly less biofilm was recovered when compared to the controls following incubation with the CFS of all 37 isolates. It is evident from the results of this study and the wider literature that under in vitro conditions, CoNS produce substances that can inhibit the growth of *S. aureus*, including drug-resistant strains. 

The effects of these naturally occurring substances could potentially be exploited to develop novel treatments for infections caused by *S. aureus*, particularly those caused by strains resistant to antibiotics. Bacteriocins produced by CoNS with activity against *S. aureus* were of interest to this study not only due to their potent targeted activity against closely related species but also due to the developing regulatory framework around these peptides through their use as preservatives and impending approval as veterinary antimicrobials. The recognition from leading regulatory bodies and the GRAS status granted to bacteriocins nisin A and colicins highlights that their development into drugs is possible [42,43]. 

WGS was performed on 11 CoNS isolates shortlisted for their ability to inhibit MRSA’s growth and biofilm formation. Although not all sequenced isolates were determined to be putative bacteriocin producers, each draft genome was found to possess multiple copies of genes encoding putative δ-lysins and PSMs, as well as for the production of quorum sensing agents, AIPs. All three are frequently identified when sequencing CoNS strains and have previously been associated with the inhibition or interference of the proliferation of *S. aureus,* making them a potential source of the activity observed [11,21,24]. While the reported role of AIPs lies in the realm of communication [20,21], the functions of δ-lysins and PSMs have yet to be fully defined. These short, amphipathic, alpha-helical peptides are produced by both coagulase-positive and negative staphylococci [44]. They play a significant role in the pathogenicity of the producing strain during infection; however, the structure of these lytic peptides also allows them to interact with the cell walls of prokaryotic cells [11]. It is through these interactions that PSMs from CoNS have been found to exert selective antimicrobial effects against a range of Gram-positive species, including *S. aureus* [11]. The draft genomes of three isolates (E4, E96 and E170) were suspected to possess incomplete bacteriocin operons, which were incapable of synthesizing the product. The presence of incomplete operons is suspected to be caused by reductive evolution, a phenomenon reported previously in the *Staphylococcus* genus [45,46]. This added to the speculation surrounding the role of δ-lysins and PSMs in the activity seen throughout the investigation. It is possible that the inhibition of MRSA was due to the production of bioactive peptides outside of the bacteriocin classification. However, given their ability to interact with human cells and potential hemolytic activity, the activity of the δ-lysins and PSMs was not investigated further.

Genome sequencing and subsequent BAGEL4.0 analysis led to the detection of two complete bacteriocin operons in the draft genomes of CoNS isolates *S. hominis* C14 and *S. epidermidis* C33. Strain C14 was originally isolated from the skin of a healthy volunteer and found to produce a nukacin family bacteriocin identical to the previously characterized nukacin KQU-13. Although other nukacin derivatives have been purified from human-isolated CoNS (e.g., nukacin KSE650 and nukacin IVK45) [39,47], this is the first reported identification of nukacin KQU-131 from a human source, with the peptide first being isolated from a strain collected from Pla-ra, a Thai fermented fish product. Nukacin ISK-1 was among the first of the nukacin-family bacteriocins to be characterized. Activity studies performed by Okuda et al. indicated purified nukacin ISK-1 has antimicrobial and anti-biofilm effects against clinical isolates of *S. aureus* (*S. aureus* MR23, *S. aureus* MR10, *S. aureus* MR11, and *S. aureus* SH1000) [48]. The initial study characterizing nukacin KQU-131 undertaken by Wilaipun et al. stated it possessed the same inhibition spectrum as that of nukacin ISK-1; however, no data was supplied in the article to support this [27]. To our knowledge, no experiments have been performed with purified nukacin KQU-131 to assess its efficacy against *S. aureus*, specifically drug-resistant strains. The spot-on-lawn assays performed with nukacin C14 demonstrated that the peptide could inhibit the growth of indicators MRSA M and MRSA T and is therefore the first application of this bacteriocin against clinical MRSA isolates. 

The *S. epidermidis* C33 draft genome analysis revealed an operon for the production of a 31 amino acid long mature epilancin family peptide. The gene encoding the core peptide was previously reported by Angelopoulou et al. [29], though it had not been purified or characterized at the time of this study. The mass of epilancin E is predicted to be 2982 Da, which is lower than that of 15X and K7, which have reported masses of 3172 Da and 3032 Da, respectively [28,49]. Although the mode of action of epilancin family bacteriocins is unknown, it is believed that the ring structure and N-terminal lactate play an important role in the activity observed. The amino acid sequence composing the regions associated with the formation of these key structures appears conserved in the mature epilancin E core peptide, and they align with those reported in the 15X and K7 peptides [28,49]. Due to the presence of these conserved regions and the high percentage identity matched epilancin E shares with 15X and epilancin K7, the mature peptide was predicted to have the same structure and inhibition spectrum. Epilancin 15X has previously been observed to express potent antimicrobial activity against drug-resistant clinical pathogens, including MRSA and vancomycin-resistant enterococci [30]. Through spot-on-lawn assays, epilancin E was observed to cause zones of clearing in the growth of the MRSA M indicator. 

The activity observed from the isolated nukacin C14 and epilancin E peptides against MRSA suggests that CoNS-derived bacteriocins could be responsible for the activity observed throughout the investigation and alludes to how their bioactivity could be exploited against human pathogens. The inhibition of MRSA aligns with the findings in the current literature, which state that CoNS-derived bacteriocins have promise against serious pathogens, and this is a cause for a deeper examination of their effects on the living system. Some bacteriocins are currently approved for human consumption in the form of natural preservatives and are applied to foods to prevent spoilage and the growth of foodborne illness-causing organisms, namely *L. monocytogenes.* However, recent developments have seen food-grade bacteriocins, purified to a pharmaceutical standard, being prepared for FDA approval against pathogens responsible for bovine mastitis [50,51]. CoNS-derived bacteriocins show inhibitory effects similar to other well-characterized peptides like nisin. Not only may staphylococcal bacteriocins act like a probiotic by stimulating host immune cells and protecting against infection [52], but bacteriocins from CoNS have been found to effectively manage infections in animal models without any perceivable harm to the host. 

When injected at a concentration of 200 mg/kg over a 48-h period, purified epidermicin NI01 was found to significantly increase the survival rates of wax moth larvae infected with *S. aureus* ATCC 11195 [53]. Topical application of epidermicin NI01 was also found to reduce or eradicate nasal carriage of MRSA in rats following administration of 0.8% solution of the peptide [15]. Lysostaphin is an endopeptidase bacteriocin produced by *Staphylococcus simulans* that has shown potent activity against *S. aureus* in vitro. Infusions of 10 µg of lysostaphin to the mammary glands of *S. aureus-infected* mice reduced the CFU/mL recovered by over 99% after only 30 min of exposure, with no cytotoxic effects on the mammary tissue [54]. Other in vivo investigations performed with non-CoNS produced bacteriocins nisin A, as well as lacticin 3147 and thuricin CD, have established that bacteriocins can be administered intraperitoneally [55], topically [56], or rectally [57] to treat an established infection or exert a protective effect against an infection taking hold. In vitro and ex vivo food and animal trials indicate that bacteriocins can work synergistically with antibiotics and other antimicrobials, in some cases lowering the dosage of antibiotics required to eradicate detectable amounts of the pathogen under investigation and inducing sensitivity within strains previously found to be resistant [58,59,60]. Safe for consumption, with little observable cytotoxicity and the ability to function in vivo, these naturally occurring substances appear to be valuable tools with the potential to support antibiotics currently in use.

Although no bacteriocin produced by a CoNS strain has received regulatory approval to date, it is evident from both this study and those performed previously [14,61] that staphylococcal-derived bacteriocins display beneficial activity which could be exploited to manage infections in a post-antibiotic era. At the conclusion of this study, it has been demonstrated that whether present in the microbiome or exploited for their bioactive products like bacteriocins, CoNS may be an effective, abundant, and accessible tool to utilize against their closely related competitor, *S. aureus*. Antimicrobial resistance is still evolving in *S. aureus;* however, the methods to counteract the species appear to have stagnated. Recognizing and investigating other naturally occurring avenues of pathogen suppression could lead to the development of probiotics or treatments with lasting effects that ensure safety and well-being in clinical settings and the wider community for years to come. Future investigations with the peptides discussed in this study will seek to characterize their activity and spectra of inhibition in order to better understand how they could best be applied as antimicrobials. 

## 4. Materials and Methods

### 4.1. Bacterial Strains

This study was undertaken as part of a larger investigation in our laboratory, which sought to study the antimicrobial capacity of CoNS isolates. Initially, 40 CoNS isolates were selected for this study; however, due to an inability to reproduce zones from two strains and a difficulty in culturing a third, 37 were carried forward for further analysis. The 37 CoNS isolates under investigation, 21 MRSA indicator strains, and one sensitive indicator strain, *M. luteus*, were obtained from the Munster Technological University Culture Collection. A second sensitive indicator strain used, *L. lactis* HP, was obtained from the University College Cork Culture Collection. The bank of 37 CoNS was originally isolated from the skin of healthy individuals or from the bloodstreams of patients with infections in Cork University Hospital. A strain of *M. luteus* known to be sensitive to antimicrobials was included in this study as an indicator to compare activity and detect strains that were potentially producing bioactive substances. The MRSA test strains were isolated from patients residing in St. James Hospital, Dublin. Sequencing revealed that the isolates in the MRSA collection were composed of five different clonal complexes (CC8, CC30, CC5, CC22, and CC45), which have previously been linked to outbreaks of disease and persistent infections in both Ireland as well as abroad [62,63,64]. A strain of *M. luteus* known to be sensitive to antimicrobials was included in this study and used as an indicator to compare activity and detect strains that were potentially producing bioactive substances. The identities of all isolates under investigation and indicator strains employed are listed in Table 3. 

### 4.2. Agar-Based Deferred Antagonism Assays to Identify CoNS Isolates with Potential Antimicrobial Activity against Methicillin-Resistant Staphylococcus aureus

To identify CoNS isolates with antimicrobial activity against MRSA, deferred antagonism assays were performed as per the protocol previously described by Lynch et al. [36]. All 37 CoNS isolates were tested against the 21 MRSA strains and the sensitive indicator, *M. luteus* CIT 3. Volumes of 200 µL of CoNS overnight cultures were transferred to individual wells of a sterile 96-well plate. TS agar was prepared and poured into square Petri dishes (120 × 120 × 17 mm^3^, Greiner, Fisher Scientific, Dublin Ireland; catalog no. 688102G), and it was allowed to set. A 96-well replicator (Boekel Scientific, PA, USA; catalog no. 140500-140501-140384) was prepared by dipping in 95% *v*/*v* ethanol and passing through a blue Bunsen flame to sterilize. Once sterilized and cooled, the pins of the replicator were dipped into the culture-containing wells of the microtiter plate and transferred to the surface of the agar. 

The spotted cultures were allowed to dry before being inverted and incubated overnight at 37 °C. Following incubation, the spot plates were placed into a CL-1000 Ultraviolet Crosslinker and UV-treated for 30 min at the maximum strength. Volumes of 20 mL of molten 0.75% *w*/*v* TS agar were prepared and inoculated with 1% of an indicator strain, i.e., MRSA or *M. luteus,* overnight culture. The inoculated agar was then poured over the UV-treated spot cultures and allowed to set. The overlaid plates were incubated for 18 h at 37 °C. Following incubation, the plates were inspected for zones of clearing in the growth of the overlaid organism, indicative of potential antimicrobial activity. Zone sizes were measured with a Vernier caliper (resolution 0.05) across the width of the zone. 

Zone sizes are presented in Table 1 in a +/− format where “−” indicates no zone of inhibition, “+” indicates a zone size of 7 mm–15 mm, “++” indicates a zone size of 7 mm–16 mm, and “+++” indicates zone sizes 16 mm–34 mm. All isolates were tested against each indicator strain in triplicate (i.e., three biological repeats). 

### 4.3. Investigating the Presence of Antimicrobial Activity in Cell-Free Supernatant

Agar-based well diffusion assays using the cell-free supernatant prepared from CoNS isolate overnight cultures were performed as per the method previously used by O’Sullivan et al. with modifications [38]. Volumes of 50 mL of TS broth were inoculated with an individual CoNS isolate and incubated overnight at 37 °C. The overnight CoNS cultures were centrifuged at 7500× *g* for 15 min (IEC CL30R, Thermo Fisher Scientific, Dublin, Ireland). Following centrifugation, the supernatant was aspirated, and filter sterilized with a 0.2 µm filter (Sarstedt, Wexford, Ireland; product no. 83.1826.001) to ensure all cells were removed. This liquid was then referred to as the cell-free supernatant. The pH of the CFS was taken using a calibrated pH meter and adjusted to pH 7 using 0.01M NaOH (Sigma Aldrich, Wicklow, Ireland; product number 655104-500G).

Volumes of 40 mL of TS agar were prepared and inoculated with 1% of an individual indicator strain, i.e., MRSA or *M. luteus*, overnight culture. The inoculated agar was then poured into square Petri dishes (120 × 120 × 17 mm^3^) and allowed to set. A glass Pasteur pipette (VWR International Limited, Derry, Ireland; catalog no. 612-3813) was sterilized by dipping in 70% *v*/*v* isopropyl alcohol and passing through a blue Bunsen flame. Once cooled, this glass pipette was used to bore wells in the inoculated agar. A sterile pipette tip was used to remove the agar plugs from the wells and discard them. Volumes of 30 µL of the CFS were transferred to individual wells in the agar. Plates were incubated upright overnight at 37 °C. Zones of inhibition in the growth of the indicator strain were measured using a Vernier caliper (resolution 0.05) across the width of the zones. All measurements presented include the diameter of the well (6 mm). All CoNS were tested against each individual indicator strain in triplicate (i.e., three biological repeats). 

### 4.4. Biofilm Formation Assay

The ability of all 21 MRSA strains to form biofilm was investigated as per the stationary microtiter plate protocol used by Mathur et al. [19]. MRSA strains were grown in 10 mL volumes of TSB + 1% (*w*/*v*) D (+) glucose (Sigma Aldrich, Wicklow, Ireland; catalog no. G8270-100G) overnight at 37 °C. Following incubation, a 1:100 dilution was performed on the MRSA overnight culture, where 100 µL of each culture was individually inoculated into 9.9 mL of fresh TSB + 1% (*w*/*v*) D (+) glucose. Each culture was vortexed before 200 µL volumes were transferred to individual wells of a 96-well, flat-bottomed, non-tissue-culture-treated plate (Sarstedt, Wexford, Ireland; product no. 82.1582.001). A sample blank was created by adding 200 µL of uninoculated TSB + 1% (*w*/*v*) D (+) glucose to individual wells of the plate. Plates were incubated under stationary conditions for 24 h at 37 °C. 

Following 24 h of incubation, the liquid was removed from the wells using a P200 pipette. Care was taken not to scrape any biofilm from the walls or bottom of wells. The wells were then rinsed with 200 µL of 1× PBS (Sigma Aldrich, Wicklow, Ireland; product no. P5244-100ML) three times and allowed to dry. An amount of 200 µL of 2% (*w*/*v*) sodium acetate (Sigma Aldrich, Wicklow, Ireland; product no. S2889-250G) was added to the wells and incubated at room temperature for 10 min. After 10 min, the sodium acetate was removed from the plate with a P200 pipette. An amount of 200 µL of 1% (*w*/*v*) crystal violet was transferred to each well and incubated at room temperature for 10 min. Following the incubation period, the crystal violet was removed from the plate with a P200 pipette. The wells were rinsed with deionized water to remove unbound crystal violet and allowed to dry. Bound crystal violet was resuspended in 95% (*v*/*v*) ethanol (Sigma Aldrich, Wicklow, Ireland; product no. 493511), and the absorbance of each well was read at 595nm using an automated plate reader (Multiskan Sky, Thermo Scientific, Dublin, Ireland). 

To assess the degree of biofilm formation, the OD of the blank (ODb) was deducted from the average of three readings taken from the strain under investigation. The resulting value (referred to here as ODc) was then assessed via the criteria established by Stepanović et al. [65], which state that ODc ≤ ODb = not a biofilm producer, ODb < ODc < 2XODb = weak biofilm producer; 2XODb < ODc ≤ 4XODb = moderate biofilm producer, and ODbX4 < ODc = strong biofilm producer. 

### 4.5. Assessing the Antimicrobial Activity of the Acid Neutralized CFS of CoNS Isolates against Strain MRSA M 

A microtiter plate growth inhibition assay, as previously described by Vijayakumar et al. with modifications, was used to determine if the CFS of CoNS isolates exerted direct antagonistic effects in aqueous media against a representative MRSA strain [66]. MRSA M was selected as the indicator strain for this experiment, as it is both a clinically relevant sequence type and was identified as a biofilm-forming strain during this study.

The CFS of each of the 37 CoNS isolates was prepared from 10 mL overnight cultures, as previously described in Section 4.3. A 1:100 dilution of MRSA overnight culture was performed by inoculating 100 µL of overnight culture into 9.9 mL of fresh double-strength TS broth. An amount of 100 µL of MRSA culture was transferred to individual wells of a 96-well, flat-bottomed, non-tissue-culture-treated plate. An amount of 100 µL of CFS was then aliquoted into the MRSA cultures, bringing the final volume in each well to 200 µL. A positive control was prepared by adding 100 µL of uninoculated TS broth to 100 µL of MRSA culture. The 96-well plate was then aseptically sealed using LightCycler^®^ 480 Sealing Foil (Roche, Sigma Aldrich, Wicklow, Ireland; ref. no. 04 729 757 001) and transferred to a Multiskan Sky automated plated reader. The plate was incubated under the following conditions: 37 °C, pulsed at OD600nm for 24 h, with readings taken at 60-min intervals (Multiskan Sky, Thermo Fisher Scientific, Dublin, Ireland). 

Each CoNS isolate was tested against the representative MRSA strain in triplicate (i.e., three biological repeats). The blank sample (uninoculated broth) was deducted from the average OD 600 nm value before they were plotted against time. 

### 4.6. Biofilm Inhibition Assay

The inhibition of MRSA biofilm formation was assessed as per the protocol used by Mathur et al. with modifications [19]. Five MRSA strains, identified as strong (V, R, and E) and moderate (M and T) biofilm formers in Section 4.4, as per the criteria established by Stepanović et al. [65], were selected for the biofilm inhibition assay. 

Selected MRSA strains were grown in 10 mL volumes of TSB + 1% (*w*/*v*) D (+) glucose at 37 °C overnight. A 1:100 dilution of the MRSA culture was performed by inoculating 100 µL into 9.9 mL of double-strength TSB + 2% glucose and vortexed. Amounts of 100 µL of the MRSA culture were added to wells of flat-bottomed, non-tissue-culture-treated 96 well plates. CFS from each of the 37 isolates was prepared from overnight cultures, as previously described in Section 4.3. An amount of 100 µL of acid-neutralized CFS from an individual CoNS isolate was added to the MRSA culture to bring the volume in each well to 200 µL. A positive control was created by combining 100 µL of MRSA 1:100 dilution with 100 µL of sterile TSB. A sample blank was created by combining 100 µL of uninoculated double-strength TSB + 2% glucose and 100 µL of sterile TSB. The 96-well plates were incubated under stationary conditions for 24 h at 37 °C. 

Following incubation, the contents were removed from the wells using a P200 pipette. Care was taken not to scrape any biofilm from the walls or bottom of wells. The wells were then rinsed with 200 µL of 1 x PBS three times and allowed to dry. An amount of 200 µL of 2% (*w*/*v*) sodium acetate was added to the wells and incubated at room temperature for 10 min. After 10 min, the sodium acetate was removed from the plate with a P200 pipette. An amount of 200 µL of 1% (*w*/*v*) crystal violet was transferred to each well and incubated at room temperature for 10 min. Following the incubation period, the crystal violet was removed from the plate with a P200 pipette. The wells were rinsed with deionized water to remove unbound crystal violet and allowed to dry. Bound crystal violet was resuspended in 95% (*v*/*v*) ethanol, and the absorbance of each well was read at 595nm using an automated plate reader. The CFS of each CoNS isolate was tested against each biofilm-forming MRSA indicator in triplicate (i.e., three biological repeats). The average absorbance reading from each CFS-treated sample minus the average absorbance of the sample blank was compared to that of the control to ascertain if there was a significant decline in biofilm formation. 

### 4.7. Whole Genome Sequencing of Selected Coagulase Negative Staphylococcus Isolates

Eleven CoNS isolates (C14, C33, E4, E6, E54, E67, E89, E96, E99, E100, and E170) were shortlisted for whole genome sequencing based on their ability to inhibit the growth and biofilm formation of MRSA. Single isolated colonies of each shortlisted CoNS isolate were individually inoculated into 200 µL of sterile 1xPBS. An amount of 100 µL of the suspension was then used to inoculate sterile TSB. The remaining 100 µL of suspension was streaked onto a sterile TSA plate. Both the inoculated broth and the plate were incubated overnight at 37 °C. Following incubation, the plate was inspected to ensure the suspension was pure. All overnight cultures were centrifuged at 4000× *g* for 20 min. The supernatant was removed before the pellet was resuspended in 500 µL cryopreservant liquid and transferred to a barcoded bead tube, both provided by Microbes NG. The beaded tube was then inverted 10 times to ensure the contents were mixed before being sent to MicrobesNG, The Biohub, Birmingham Research Park, 97 Vincent Drive, Birmingham, United Kingdom, for whole genome sequencing. 

The annotated draft genomes were received in FASTA format and as GBK and GFF files. Assembled contigs were visualized with the genome viewer Artemis (https://www.sanger.ac.uk/tool/Artemis/; accessed on 22 June 2021). Analysis of the draft genomes for potential antimicrobial encoding genes was performed using online genome mining tools antiSMASH 5.0 (https://antismash.secondarymetabolites.org/, accessed on 23 June 2021) and BAGEL4.0 (http://bagel4.molgenrug.nl/index.php, accessed on 23 June 2021) using the default parameters. Comparison to previously characterized bacteriocins and prediction of potential gene function was performed using the bacteriocin database Bactibase (http://bactibase.hammamilab.org/about.php, accessed on 24 November 2021) and the NCBI repository and search tool BLASTp (https://blast.ncbi.nlm.nih.gov/Blast.cgi, accessed on 05 February 2024). The species identity of each isolate was predicted by performing alignments of the nucleotide sequences with the 16S rRNA genes strains of the same species present on the NCBI RefSeq Database (NCBI RefSeq Targeted Loci Project (nih.gov)). A percentage identity match of 97% was the cutoff criterion for considering isolates to be the same species in this study. 

### 4.8. Analysis of Putative Bacteriocin Clusters Identified in CoNS Isolates

Analysis of the putative bacteriocin operons detected in the genomes of the CoNS isolates through BAGEL4.0 and antiSMASH was performed using the NCBI Conserved Domain search tool (https://www.ncbi.nlm.nih.gov/Structure/cdd/wrpsb.cgi 21 February 2022), alignment tools ebi EMBOSS NEEDLE (https://www.ebi.ac.uk/Tools/psa/emboss_needle/ 10 January 2022) and Jalviewer (https://www.jalview.org/ 21 February 2022), and biochemical and physicochemical characteristic analysis programs CLUSTALW MEGA11 (https://www.megasoftware.net/ 21 February 2022). A single amino acid change in the peptide sequence was considered a novel variant, whereas a less than 30% identity match to the amino acid sequences of previously characterized or previously reported bacteriocins was the criteria that a gene may encode a novel peptide. 

### 4.9. Purification of Identified Bacteriocins and Investigation of Antimicrobial Activity 

Purification of the putative nukacin C14 peptide from the CFS of strain *S. hominis* C14 was undertaken using a method previously used by Field et al. for bacteriocin isolation [67]. Here, 1 L of overnight culture was centrifuged at 7000× *g* for 15 min using a Sorvall RC 6+ model centrifuge. The supernatant was retained, and the cell pellet was discarded. The supernatant was filtered through a 30 cm long column, with an internal diameter of 2.5 cm, packed with 60 g of Amberlite XAD-16 beads (Sigma Aldrich, Wicklow, Ireland) and prewashed with 1 L of sterile water. After passing the supernatant through the column, followed by washing with 500 mL 30% (*w*/*v*) ethanol, bound nukacin was eluted using 400 mL 70% 2-propanol 0.1% trifluoroacetic acid. 

This elution was transferred to a rotary evaporator (Buchi, Flawil, Switzerland). The 2-propanol was removed, concentrating the volume of approximately 300mL. The sample was then applied to a 10 g (60 mL) SPE C-18 Bond Elute Column (Phenomenex, Cheshire, UK), pre-equilibrated with 60 mL methanol and 60 mL water. An amount of 120 mL of 30% (*v*/*v*) ethanol was washed through the column prior to the peptide being eluted in 60 mL of 70% 2-propanol 0.1% trifluoracetic acid (TFA). This 60 mL was separated into 15 mL samples, which were subjected to further rotary evaporation to reduce each fraction to a volume of 2 mL and run through a Phenomenex (Phenomenex, Cheshire, UK), C12 reverse phase (RP)-HPLC column (Jupiter 4u Proteo 90 Å, 250 × 10.0 mm, 4 μm), equilibrated with 25% 2-propanol and 0.1% TFA. The sample moved through the column at a flow rate of 3.2 mL per minute. All fractions were collected and plated against the sensitive indicator, *M. luteus,* using a WDA as previously described in Section 4.2. Following incubation, the plates were inspected for zones of clearing in the growth of the indicator, indicative of antimicrobial activity. 

Fractions determined to possess bioactivity were pooled and then placed in the rotary evaporator to remove any remaining acetonitrile. A second HPLC separation was performed using the pooled fractions under the previously discussed conditions to concentrate the peptide further. MALDI TOF Mass Spectrometric (MS) analysis was performed on the active fractions as per the method used by Field et al. [67] to detect the presence of a mass peak correlating with that previously determined for nukacin KQU-131 (3004Da) given that nukacin C14 was predicted to have an identical mass. Active fractions with a peak at 3004 Da were pooled and concentrated. The concentrated, pooled fractions were tested using spot assays, whereby overnight cultures of representative MRSA indicators MRSA M and MRSA T were adjusted to 0.5 McFarland standard and swabbed evenly over the entire surface of fresh TSA plates. Volumes of 20 µL of the active fractions were spotted on the surface of the plate and allowed to dry before being incubated overnight at 37 °C. Zones of clearing in the growth of the indicator were considered indicative of antimicrobial activity. 

The protocol used for the isolation of nukacin C14 was found to be inappropriate for the purification of the epilancin variant as activity appeared to be lost from the XA 16N and C18 columns following washing with ethanol. To successfully extract the peptide from the CFS of *S. epidermidis* C33 culture, the following adjusted method was used. *S. epidermidis* C33 was grown overnight in a shaking incubator at 180 rpm, at 37 °C, in 5 mL of BHI broth (Sigma Aldrich, Wicklow, Ireland; product no. 75917). Following incubation, the culture was used to create a 1% inoculum in 100 mL of fresh BHI broth. The 100 mL C33 inoculum was incubated for 22 h at 37 °C, shaking at 180 rpm. The cells were separated from the supernatant by centrifugation at 4400 rpm for 20 min at room temperature. An amount of 100 mL of supernatant was applied to a 12 mL, 2 g Strata^®^ C18-E SPE column (Phenomenex, Cheshire, UK) pre-equilibrated with methanol and water. The column was washed with 18 mL 25% (*v*/*v*) ethanol and then 12 mL 70% 2-propanol and 0.1% TFA. To ensure that activity was not lost following passage through the column, the eluted material from both the wash step and 70% 2-propanol and 0.1% TFA elution were tested using WDAs against sensitive indicator, *L. lactis* HP.

As activity was not found to be lost, the 2-propanol was removed from the sample using rotary evaporation before being applied to a Proteo Jupiter C12 RP-HPLC column (250 × 10 mm^2^, 4 µ, 90 Å) running a 20–50% acetonitrile and 0.1% TFA gradient where buffer A is 0.1% TFA and B is 100% acetonitrile and 0.1% TFA. HPLC was performed on a SIL-10AP Shimadzu autosampler system (Shimadzu, Duisburg, Germany). The eluent was monitored at 214 nm, and fractions were collected at approximately 25-s intervals. All fractions collected were assayed using WDAs against *L. lactis* HP, and active fractions checked for masses of interest using MALDI TOF MS. The potentially active fractions found to share a mass peak correlating with the predicted mass of the epilancin variant (epilancin E) were pooled and then placed in the rotary evaporator to remove any remaining acetonitrile. Another HPLC separation was performed under the same conditions using the pooled fractions in order to concentrate the peptide further. The pooled and concentrated fractions confirmed through MALDI TOF MS to have a peak correlating with the mass of epilancin E were tested through spot assays against MRSA M and MRSA T, as described above. 

## 5. Conclusions

This study investigated the ability of human-derived CoNS to inhibit clinical isolates of MRSA, a difficult-to-treat cause of infection routinely identified in clinical settings. At its conclusion, the study confirmed through several tests that the CFS samples prepared from overnight cultures of the CoNS isolates could prevent the growth and biofilm formation of the MRSA indicators they were tested against. These tests suggested that substances with anti-MRSA activity were being produced. As the primary focus of our laboratory is the identification of bacteriocins with clinically relevant activity, WGS was then used to locate operons encoding peptides that may have caused the activity observed. The identification of nukacin C14 and epilancin E affirmed that bacteriocins produced by CoNS can have activity against MRSA, as both were found to inhibit MRSA M through spot-on-lawn assays. Going forward, CoNS presents itself as an abundant and accessible tool to utilize against drug-resistant *S. aureus*, either through direct competition in the commensal flora or the production of potent, naturally occurring antimicrobials that are effective against serious human pathogens. 

## Figures and Tables

**Figure 1 antibiotics-13-00338-f001:**
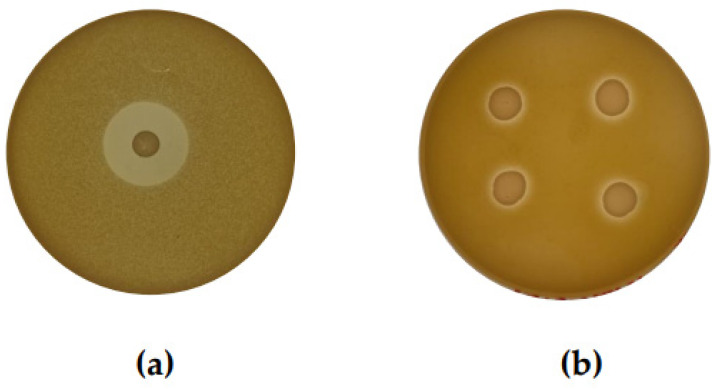
(**a**) Representative CoNS isolate C14 overlaid with (**a**) sensitive indicator, *M. luteus.* (**b**) Biological repeats of representative CoNS isolate C14 overlaid with indicator strain MRSA R, an antibiotic-resistant *S. aureus* strain isolated from a clinical setting. Deferred antagonism assays were performed to assess the degree of bioactivity within a bank of human-isolated CoNS and identify isolates that may have inhibitory effects against MRSA.

**Figure 2 antibiotics-13-00338-f002:**
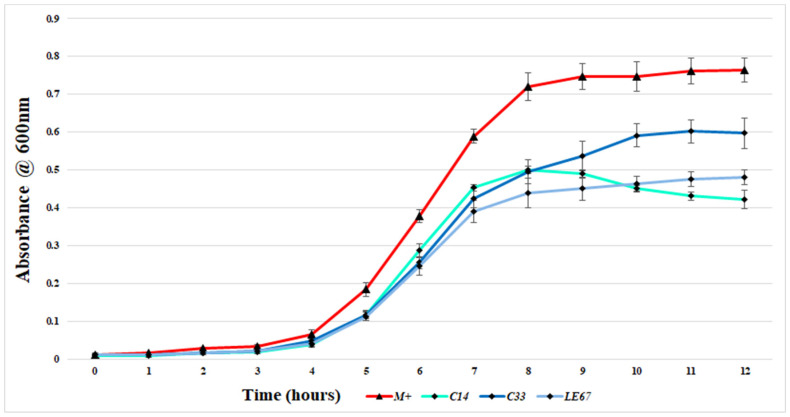
Hourly absorbance readings of the growth of representative MRSA indicator strain, MRSA M, cultured in the presence of the acid-neutralized CFS prepared from three representative CoNS overnight cultures. The control and test cultures were incubated at 37 °C with continuous pulsing. Absorbances taken every 60 min over a 12-h period at 600 nm. Plotted points are the average ± SD of three biological repeats minus the sample blank (uninoculated broth). A positive control was created by incubating MRSA M (represented by M+ in the figure) in the absence of CFS.

**Figure 3 antibiotics-13-00338-f003:**
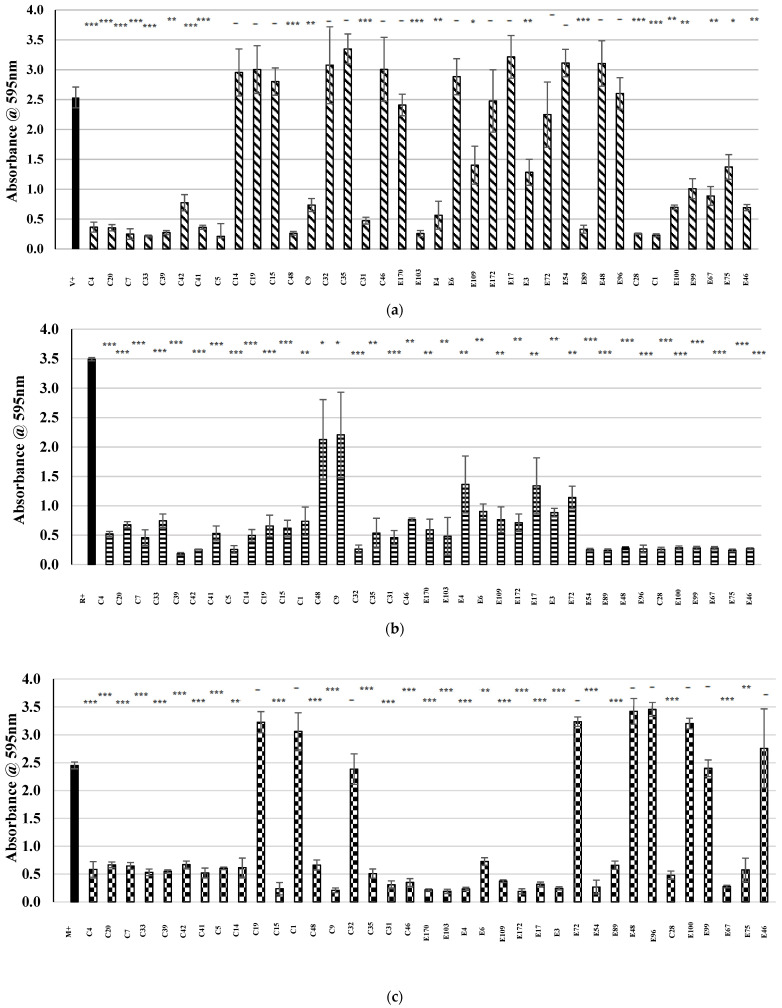
The effects of the acid-neutralized CFS derived from the overnight cultures of CoNS isolates on biofilm formation by five clinical isolates of MRSA. The results are presented in the following ways: (
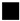
) represents the positive control in each assay (MRSA cultured in the absence of CFS), and patterned bars represent an MRSA strain cultured in the presence of CFS. The effect of the CFS was investigated against 5 MRSA indicators (**a**) V, (**b**) R, (**c**) M, (**d**) T, and (**e**) E. The MRSA strains were cultured for 24 h at 37 °C under stationary conditions in the presence of 100 µL of acid-neutralized CFS. The absorbance of MRSA cultured in the presence of CFS was compared to the positive control. Asterixis indicates a significant difference in biofilm formed when compared to the control, where *** denotes a reduction of *p* < 0.001, ** denotes a reduction of *p* < 0.01, and * denotes a reduction of *p* < 0.05. In contrast, “-” denotes a non-statistically significant reduction, no change, or an increase in biofilm formation.

**Figure 4 antibiotics-13-00338-f004:**
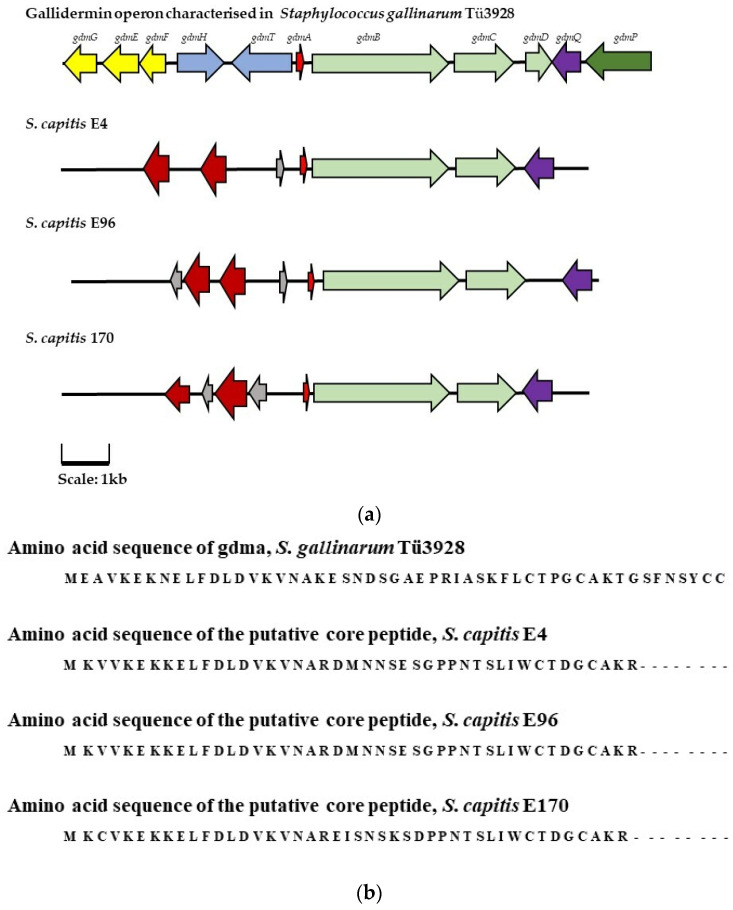
(**a**) The characterized 11 gene operon for the biosynthesis of gallidermin compared to the gallidermin-associated AOIs detected in the genomes of strains E4, E96, and E170. (
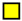
) represents the genes encoding immunity peptides *gdmGFE*, (
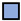
) represents the genes encoding transport/export proteins *gdmHT*, (
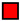
) represents the gene encoding the core pre-peptide, (
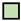
) represents modification genes *gdmBCD* (
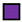
) represents regulation/activator gene *gdmQ,* and (
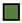
) represents processing gene *gdmP*. Genes with a corresponding function in the AOIs identified in isolates E4, E96, and E170 are highlighted in the same color. In the putative bacteriocin operons identified in isolates E4, E96, and E170 (
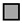
) represents hypothetical proteins, and (
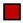
) represents putative lipoproteins. (**b**) The amino acid sequences encoding the gallidermin pre-peptide (*gdmA*) and the presumptive core peptides identified in the AOIs detected in isolate E4, E96, and E170 by BAGEL4.0.

**Figure 5 antibiotics-13-00338-f005:**
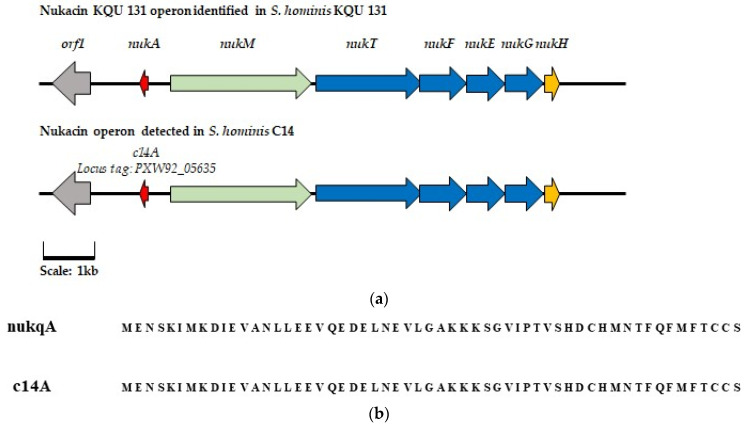
(**a**) The characterized eight-gene operon for the biosynthesis of nukacin KQU-131 compared to the AOI identified in *S. hominis* C 14 through genomic analysis performed with BAGEL4.0. (
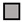
) represents regulation gene *orf*1 (
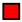
) represents the core pre-peptide gene (
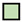
) represents modification peptide *nukM* (
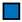
) represents transport and permease genes *nukTFEG*, and (
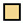
) represents hypothetical protein *nukH*. Genes with a corresponding function in the AOI detected in *S. hominis* C14 are highlighted in the same color. (**b**) Alignment of the amino acid sequence of the nukacin KQU-131 pre-peptide and that of the putative core peptide sequence identified in *S. hominis* C14.

**Figure 6 antibiotics-13-00338-f006:**
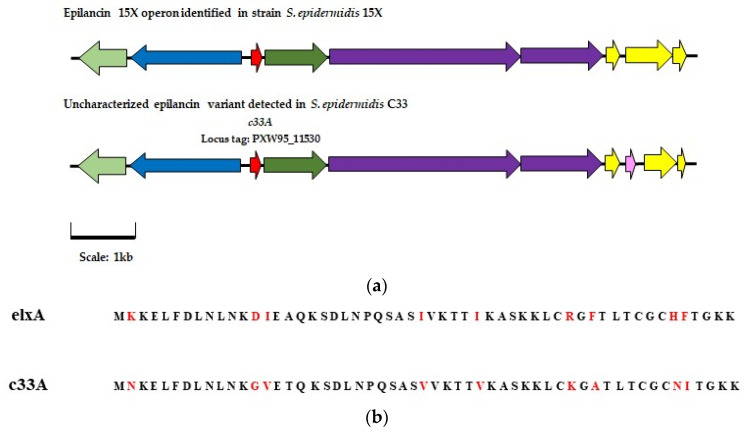
(**a**) The characterized operon for the biosynthesis of derivative epilancin 15X compared to the AOI identified in isolate C 33. (
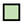
) represents modification gene elxO (
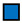
) represents transport gene exlT (
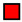
) represents the gene for the core peptide gene exlA (
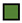
) represents processing gene exlP (
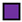
) represents bacteriocin biosynthesis gene elxBC, and (
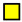
) represents immunity genes exlI(1/2/3). Genes with a corresponding function in the C 33 AOI are highlighted in the same color. In the AOI identified in isolate C33 (
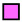
) represents a gene predicted to encode a putative intermembrane-associated protein, potentially involved with bacteriocin immunity. (**b**) Alignment of the amino acid sequence of the gene encoding the epilancin 15X peptide and that of the gene encoding the putative core peptide detected in isolate C33, where the deviations in the amino acid sequences are highlighted in red.

**Figure 7 antibiotics-13-00338-f007:**
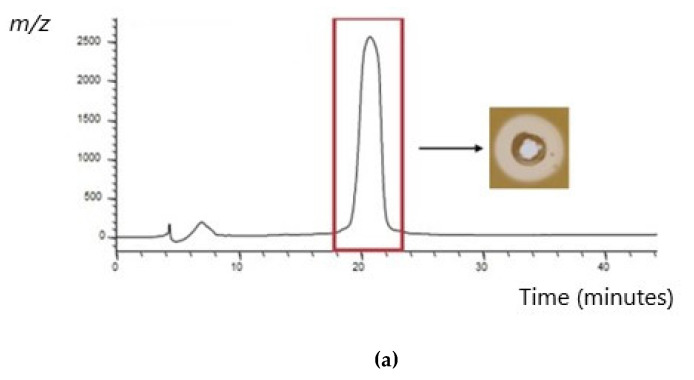
(**a**) Reversed–phase high performance liquid chromatography (RP-HPLC) profile for the purification of nukacin C14 using a Phenomenex C12 reverse-phase column. Bioactivity was determined through WDAs where *M. luteus* was used as an indicator strain. The presence of a zone of clearing indicated activity was present in two separate fractions, highlighted by a red box. (**b**) MALDI-TOF MS analysis of the active fractions revealed a mass of 3004 Da, which corresponds with the previously determined mass of nukacin KQU 131, to which nukacin C14 is identical.

**Figure 8 antibiotics-13-00338-f008:**
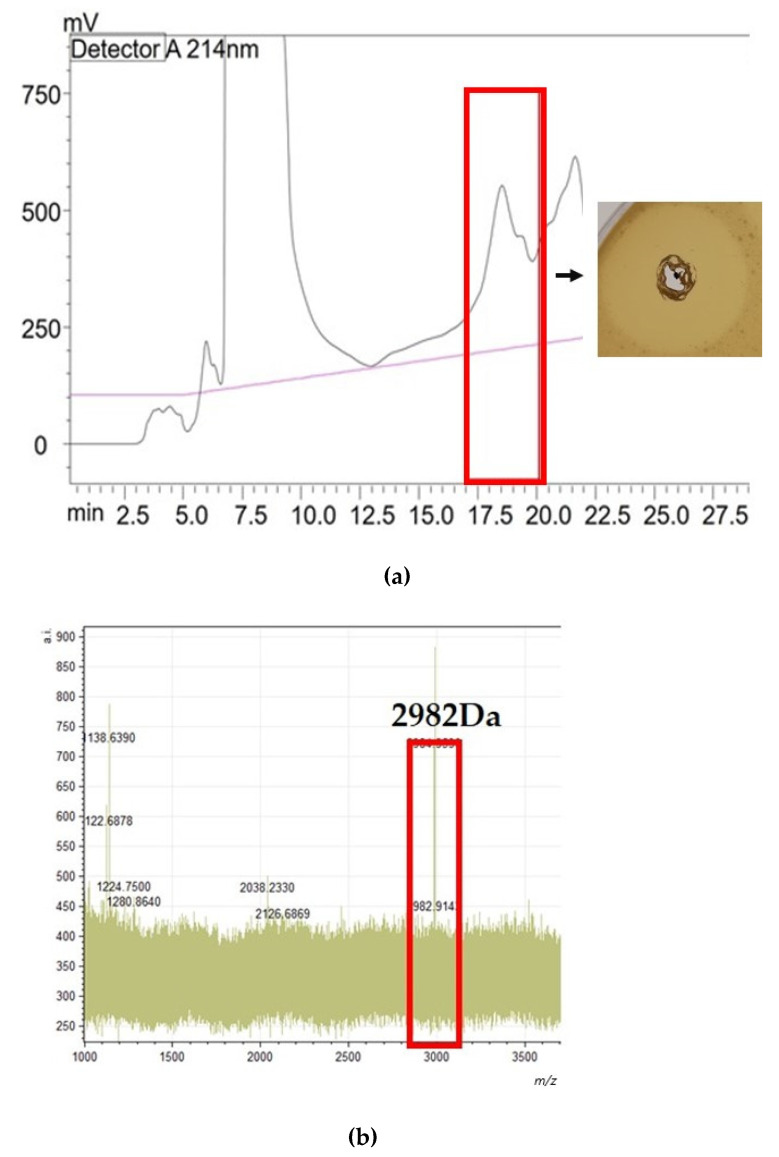
(**a**) HPLC profile obtained from the purification of epilancin E from 100 mL of CFS prepared from *S. epidermidis* C33 overnight culture. Bioactivity was determined through WDAs where *L. lactis* was used as an indicator strain. The presence of a zone of clearing indicated activity was present in eight separate fractions, highlighted by a red box. (**b**) MALDI TOF MS presented a mass of approximately 2982 Da in the active fractions. This mass peak corresponds with that calculated for epilancin E.

**Figure 9 antibiotics-13-00338-f009:**
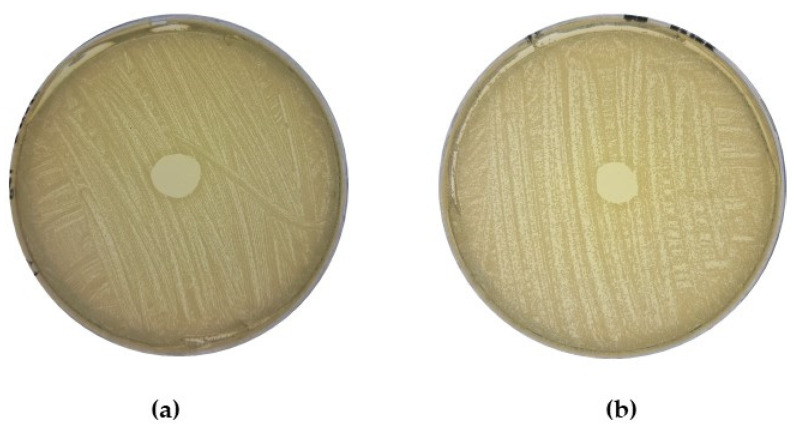
Spot-on-lawn assays were performed with the isolated (**a**) nukacin C14 peptide and the isolated (**b**) epilancin E peptide against representative MRSA indicator strain, MRSA M. For spot-on-lawn assays, the concentration of the chosen indicator was standardized and swabbed over the entire surface of a fresh TSA plate. An amount of 20 µL of the representative peptide was spotted onto the plate and allowed to dry before being incubated overnight. Zones of clearing in the growth of the target strain were indicative of antimicrobial activity.

**Table 1 antibiotics-13-00338-t001:** Agar-based deferred antagonism assays for the identification of CoNS isolates with potential antimicrobial activity against sensitive indicator *M. luteus* (ML) and clinically isolated MRSA strains (B–W). Results are presented in a +/− format, where “−” indicates no zone of inhibition, “+” indicates a zone size of ≤7 mm–7 mm, “++” indicates a zone size of 8 mm–15 mm, and “+++” indicates zone sizes 16 mm–34 mm.

Indicator Strain.																						
CoNS Isolate	ML	B	C	D	E	F	G	H	I	J	K	L	M	N	O	P	R	S	T	U	V	W
C1	++	−	−	−	−	−	−	−	−	−	−	−	−	−	−	−	−	−	−	−	−	−
C4	+	−	−	−	−	−	−	−	−	+	−	−	−	−	−	−	−	−	−	−	−	−
C5	+	−	−	−	−	−	−	−	−	−	+	+	−	−	−	−	+	−	−	−	−	−
C7	++	−	−	−	−	−	−	−	−	−	−	−	−	−	−	−	−	−	−	−	−	−
C9	++	−	−	−	−	−	−	−	−	+	−	−	−	−	−	−	−	−	−	−	−	−
C14	+++	−	−	−	−	−	+	−	−	−	−	−	−	−	−	−	+	−	−	−	−	−
C15	+++	+	−	−	−	+	−	−	−	−	−	−	−	−	−	−	−	−	−	−	−	−
C19	++	−	−	−	−	−	−	−	−	−	−	−	−	−	−	−	−	−	−	−	−	−
C20	+	−	−	−	−	−	−	−	−	−	−	−	−	−	−	−	−	−	−	−	−	−
C31	++	−	−	−	−	−	−	−	−	−	−	−	−	−	−	−	−	−	−	−	−	−
C32	++	−	−	−	−	−	−	−	−	−	−	−	−	−	−	−	−	−	−	−	−	−
C33	+++	−	−	−	−	−	−	−	−	−	−	−	−	−	−	−	−	−	−	−	−	−
C35	++	−	−	−	−	−	−	−	−	−	−	−	−	−	−	−	−	−	−	−	−	−
C39	++	−	−	−	−	−	−	−	−	−	−	−	−	−	−	−	−	−	−	−	−	−
C41	+	−	−	−	−	−	−	−	−	−	−	−	−	−	−	−	−	−	−	−	−	−
C42	++	−	−	−	−	−	−	−	−	−	−	−	−	−	−	−	−	−	−	−	−	−
C46	++	−	−	−	−	−	−	−	−	−	−	−	−	−	−	−	−	−	−	−	−	−
C48	+	−	−	−	−	−	−	−	−	−	−	−	−	−	−	−	−	−	−	−	−	−
C28	+	−	−	−	−	−	−	−	−	−	−	−	−	−	−	−	−	−	−	−	−	−
E3	++	−	−	−	−	−	−	−	−	−	−	−	−	−	−	−	−	−	−	−	−	−
E4	+	−	−	−	−	−	−	−	−	−	−	−	−	−	−	−	−	−	−	−	−	−
E6	++	−	−	−	−	−	−	−	−	−	−	−	−	−	−	−	−	−	−	−	−	−
E17	+	−	−	−	−	−	−	−	−	−	−	−	−	−	−	−	−	−	−	−	−	−
E46	++	−	−	−	−	−	−	−	−	−	−	−	−	−	−	−	−	−	−	−	−	−
E48	+	−	−	−	−	−	−	−	−	−	−	−	−	−	−	−	−	−	−	−	−	−
E54	+	−	−	−	+	−	−	−	−	−	−	−	−	−	−	−	−	−	−	−	−	−
E67	++	−	+	+	+	−	+	+	+	−	+	+	+	+	−	+	+	−	+	+	+	+
E72	+	−	−	−	−	−	−	−	−	−	−	−	−	−	−	−	−	−	−	−	−	−
E75	+	−	−	−	−	−	−	−	−	−	−	−	−	−	−	−	−	−	−	−	−	−
E89	+	−	−	−	−	−	−	−	−	−	−	−	−	−	−	−	−	−	−	−	−	−
E96	+	−	−	−	−	−	−	−	−	−	−	−	−	−	−	−	−	−	−	−	−	−
E99	+	−	−	−	−	−	−	−	−	−	−	−	−	−	−	−	−	−	−	−	−	−
E100	+	−	−	−	−	−	−	−	−	−	−	−	−	−	−	−	−	−	−	−	−	−
E103	++	−	−	−	−	−	−	−	−	−	−	−	−	−	−	−	−	−	−	−	−	−
E109	+	+	−	−	−	−	−	−	−	−	−	−	−	−	−	−	−	−	−	−	−	−
E170	+	−	−	−	−	−	−	−	−	−	−	−	−	−	−	−	−	−	−	−	−	+
E172	+	−	−	−	−	−	−	−	−	−	−	−	−	−	−	−	−	−	−	−	−	−

**Table 2 antibiotics-13-00338-t002:** Areas of interest detected by BAGEL4.0 and identified through analysis with Artemis in the genomes of the sequenced CoNS isolates. Areas of interest potentially responsible for the biosynthesis of bacteriocins are highlighted in red.

CoNS Isolate	CoNS Isolate Identity	Areas of Interest Detected by BAGEL4.0 and Artemis Analysis
C14	*S. hominis*	Autoinducing peptide operon δ-lysin/PSM operons Bacteriocin J46 encoding operon
C33	*S. epidermidis*	Autoinducing peptide operon δ-lysin/PSM operons Epilancin 15X encoding operon
E4	*S. capitis*	Autoinducing peptide operon δ-lysin/PSM operons Gallidermin-associated operon
E6	*S. epidermidis*	Autoinducing peptide operon δ-lysin/PSM operons
E54	*S. epidermidis*	Autoinducing peptide operon δ-lysin/PSM operons
E67	*S. epidermidis*	Autoinducing peptide operon δ-lysin/PSM operons
E89	*S. epidermidis*	Putative autoinducing peptide operon δ-lysins and PSM
E96	*S. capitis*	Autoinducing peptide-associated operon δ-lysin/PSM operons Gallidermin-associated operon
E99	*S. epidermidis*	Putative autoinducing peptide operon δ-lysin/PSM operons
E100	*S. epidermidis*	Putative autoinducing peptide operon δ-lysin/PSM operons
E170	*S. capitis*	Autoinducing peptide-associated operon δ-lysin/PSM operons Gallidermin-associated operon

**Table 3 antibiotics-13-00338-t003:** Identity of CoNS isolates and the indicator strains used and investigated in this study. CoNS isolate identity was determined through partial 16S rRNA gene sequencing and MALDI TOF mass spectrometry. CoNS isolates, indicator strains, MRSA and *M. luteus* were grown in Tryptone Soy (TS) broth and on TS agar at 37 °C for 18 h unless otherwise specified. The *L. lactis* indicator strain was grown in M17 media supplemented with 0.5% glucose at 30 °C for 16 h.

Isolate or Indicator Name in This Study	Isolate Identity
C1	*S. epidermidis*
C4	*S. hominis*
C5	*S. hominis*
C7	*S. epidermidis*
C9	*S. epidermidis*
C14	*S. hominis*
C15	*S. epidermidis*
C19	*S. epidermidis*
C20	*S. epidermidis*
C31	*S. epidermidis*
C32	*S. lugdunensis*
C33	*S. epidermidis*
C35	*S. epidermidis*
C39	*S. warneri*
C41	*S. capitis*
C42	*S. epidermidis*
C46	*S. epidermidis*
C48	*S. epidermidis*
C28	*S. capitis*
E3	*S. epidermidis*
E4	*S. capitis*
E6	*S. epidermidis*
E17	*S. epidermidis*
E46	*S. epidermidis*
E48	*S. epidermidis*
E54	*S. epidermidis*
E67	*S. epidermidis*
E72	*S. epidermidis*
E75	*S. epidermidis*
E89	*S. epidermidis*
E96	*S. capitis*
E99	*S. epidermidis*
E100	*S. epidermidis*
E103	*S. epidermidis*
E109	*S. epidermidis*
E170	*S. capitis*
E172	*S. epidermidis*
MRSA B	*S. aureus* 0.0066 (IIIv) ST239
MRSA C	*S. aureus* 0.1206 (Iv) ST250
MRSA D	*S. aureus* 0.1239 (III) ST239
MRSA E	*S. aureus* 0.1345 (llv) ST8
MRSA F	*S. aureus* 0073 (III) ST239
MRSA G	*S. aureus* 0104 (III) ST239
MRSA H	*S. aureus* 0220 (ll) ST5
MRSA I	*S. aureus* 0242 (Iv) ST30
MRSA J	*S. aureus* 0308 (1A) ST247
MRSA K	*S. aureus* 3045 (llv) ST8
MRSA L	*S. aureus* 3144 (llv) ST8
MRSA M	*S. aureus* 3488 (lvv) ST8
MRSA N	*S. aureus* 3581 (IA) ST247
MRSA O	*S. aureus* 3594 (II) ST36
MRSA P	*S. aureus* 3596 (llv) ST8
MRSA R	*S. aureus* E1038 (IIV) ST8
MRSA S	*S. aureus* E1139 (IV) ST45
MRSA T	*S. aureus* E1174 (Iv) ST22
MRSA U	*S. aureus* E1185 (Iv) ST12
MRSA V	*S. aureus* E1202 (II) ST496
MRSA W	*S. aureus* M03/0073 (III) ST239
*M. luteus* CIT3	*M. luteus*
*L. lactis* HP	*L. lactis*

## Data Availability

The draft genomes of the sequenced coagulase negative staphylococcal isolates have been deposited in GenBank under bioproject number PRJNA938073.

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
