# Peer review of "Inhibition of Clinical MRSA Isolates by Coagulase Negative Staphylococci of Human Origin"

_antibiotics, 2024, doi:10.3390/antibiotics13040338_

Round 1

Reviewer 1 Report

Comments and Suggestions for Authors

It is a novel and interesting study on the ability of CONs strains to inhibit the growth of methicillin-resistant S. aureus  (MRSA) and the prospective role CoNS may play as a source of bioactive 28 substances with activity against critical pathogens

The summary reflects what is indicated later in the article.

The introduction is extensive and well documented.

Although the journal indicates that the article can be sent in a free format, I consider that material and methods should always come before the results and discussion. I think this should be changed.

For example, when it indicates …..DAAs were then used to examine all 37 isolates for antimicrobial activity against 21 MRSA strains provided by the 84 MTU Culture Collection, originally obtained from patients with infections. Nine CoNS isolates, (C4, C5, C9, C14, C15, 85 E54, E67, E109 and E170),  its precedence is not clear to me, I would have to read it at the end of the work and this does not seem logical to me. And so with other results….

The conclusions section is very broad and repetitive of the discussion. I think it should be simplified into a few short sentences.

Author Response

We thank this reviewer for their positive comments on our manuscript. 

  • Although the journal indicates that the article can be sent in a free format, I consider that material and methods should always come before the results and discussion. I think this should be changed.

For example, when it indicates …..DAAs were then used to examine all 37 isolates for antimicrobial activity against 21 MRSA strains provided by the 84 MTU Culture Collection, originally obtained from patients with infections. Nine CoNS isolates, (C4, C5, C9, C14, C15, 85 E54, E67, E109 and E170),  its precedence is not clear to me, I would have to read it at the end of the work and this does not seem logical to me. And so with other results….

We agree with the reviewer that is not always the convention to have the methods section after the discussion, however this was the format of the MDPI Antibiotics manuscript template that we used so is therefore in keeping with the requirements set for publication.

  • The conclusions section is very broad and repetitive of the discussion. I think it should be simplified into a few short sentences.

We agree the conclusion bore a lot of similarities to the discussion and have shortened it to the key elements. It now reads: [710-721]

“This study investigated the ability of human derived CoNS to inhibit clinical isolates of MRSA, a pathogen routinely isolated from infections in nosocomial settings. At its conclusion, this study confirmed the growth and biofilm formation of the MRSA indicators used were negatively impacted by bioactive substances produced by the commensal CoNS strains. Bacteriocins nukacin C14 and epilancin E which were purified from the supernatants of CoNS isolates S. hominis C14 and S. epidermidis C33 respectively may have contributed to the inhibitory effects observed. Going forward, CoNS present themselves as an abundant and accessible tool to utilize against drug resistant S. aureus, either through direct competition in the commensal flora, or via the production of potent naturally occurring antimicrobials”.

We once again would like to extend our thanks to the reviewers for their time, and we hope that the revised manuscript (please see attached) is now suitable for publication.

Reviewer 2 Report

Comments and Suggestions for Authors

Dear authors,

Thank you for your submission. The topic of inhibition of clinical MRSA isolates by coagulase-negative 2 staphylococci of human origin is very interesting.

All methods in this research are sufficient in detail with appropriate statistical tests. However, some points need to be clarified.

I would recommend a minor revision as follows:

Point 1: Please carefully check the manuscript format throughout the manuscript.

Point 2: Table 1 should be reformatted to landscape.

Point 3: Figure 2 should be reformatted to landscape.

Point 4: Figure 3 needs more quality as it is hard to read.

Point 5: In table 2, the font can be bigger.

Author Response

We thank this reviewer for their positive comments on our manuscript.

1) Point 1: Please carefully check the manuscript format throughout the manuscript.

We have reviewed the format again and have made the following adjustments:

  • Throughout the manuscript the text has been checked to ensure that it is Palatino linotype font size 11, spacing before and after paragraphs has been added, and that all text follows the appropriate margins, as per the MDPI template.

2) Point 2: Table 1 should be reformatted to landscape.

Table 1 and the page it is on (page 4) are currently in landscape format. We would like to accommodate the reviewer’s comments but require clarification on what adjustments could be made.

We have extended the width of Table 1 to better fill the space, but are happy to adjust it further.

3) Point 3: Figure 2 should be reformatted to landscape.

As with point 2, Figure 2 and the page it is on (page 6) are currently in landscape. If this comment could be detailed further, we would be happy to make the changes the reviewer suggests.

4) Point 4: Figure 3 needs more quality as it is hard to read.

We apologise for the poor quality of the images and hope the amended figures are of a better standard. These changes were made on page 8 and 9.

5) Point 5: In table 2, the font can be bigger.

To improve legibility, the font size in Table 2 (page 11) was increase. We hope this size is suitable.

We once again would like to extend our thanks to the reviewers for their time, and we hope that the revised manuscript is now suitable for publication. Please see attachment. 

Reviewer 3 Report

Comments and Suggestions for Authors

This study screened 37 Coagulase-negative staphylococci (CoNS) for their ability to inhibit methicillin-resistant Staphylococcus aureus (MRSA) growth. Eleven CoNS isolates were identified with inhibitory effects against MRSA, two of which possessed complete putative bacteriocin operons encoding nukacin and a novel epilancin variant. Subsequent isolation and identification of these peptides from Staphylococcus hominis C14 and Staphylococcus epidermidis C33 cultures confirmed their ability to inhibit MRSA growth, highlighting the potential of CoNS as sources of bioactive substances against antibiotic-resistant pathogens.

Some minor corrections and justification are required

1. The word *In vitro* should be in italic in lines 451, 521, and 526.

2. Where were these MRSA strains procured from?

3. Were these MRSA strains handled in a Biosafety Level 3 (BSL-3) facility?

4. Which HPLC system was used? Please include the company name.

--------------

Comments on the Quality of English Language

Minor editing of English language required

Author Response

We thank this reviewer for their comments on our manuscript.

  1. The word *In vitro* should be in italic in lines 451, 521, and 526.

This has been amended in the revised manuscript - lines 75, 425, 495, and 500.

  1. Where were these MRSA strains procured from?

The MRSA strains were provided by the culture collection at St. James’ Hospital, Dublin, and were originally isolated from patients on site. The strains were acquired for use in a different study in our laboratory where they were characterised. Permission was sought for use of 21 MRSA strains as indicators in this study. Strain details are provided in the methods section (4.1. Bacterial Strains), including source, the culture collections which they were housed at, the sequence typing and clonal complex of each indicator.

  1. Were these MRSA strains handled in a Biosafety Level 3 (BSL-3) facility?

MRSA strains were handled in a BSL-2 facility which adheres to all national safety standards for working with a variety of pathogens (Biological Agents Code of Practice – Health and Safety Authority).

  1. Which HPLC system was used? Please include the company name.

The HPLC system used was a SIL-10AP Shimadzu autosampler. This detail has now been included in section 4.10 Purification of identified bacteriocins and investigation of antimicrobial activity. Line 761 now reads “As activity was not found to be lost, the 2-propanol was removed from the sample using rotary evaporation before being applied to a Proteo Jupiter C12 RP-HPLC column (250 x 10 mm, 4µ, 90Å) running a 20-50% acetonitrile and 0.1% TFA gradient where buffer A is 0.1% TFA and B is 100% acetonitrile and 0.1% TFA. HPLC was performed on a SIL-10AP Shimadzu autosampler system.”

We once again would like to extend our thanks to the reviewers for their time, and we hope that the revised manuscript is now suitable for publication. Please see attachment.
